# Environmental Dispersion of Multiresistant *Enterobacteriaceae* in Aquatic Ecosystems in an Area of Spain with a High Density of Pig Farming

**DOI:** 10.3390/antibiotics14080753

**Published:** 2025-07-25

**Authors:** Javier Díez de los Ríos, Noemí Párraga-Niño, María Navarro, Judit Serra-Pladevall, Anna Vilamala, Elisenda Arqué, María Baldà, Tamar Nerea Blanco, Luisa Pedro-Botet, Óscar Mascaró, Esteban Reynaga

**Affiliations:** 1Internal Medicine Department, Multidisciplinary Inflammation Research Group, Hospital Universitari de Vic, 08500 Vic, Spain; 2Fundació Lluita Contra les Infeccions, Infectious Diseases Department, Hospital Universitari Germans Trias i Pujol, 08916 Badalona, Spain; 3Escola de Doctorat, Universitat de Vic–Universitat Central de Catalunya (UVIC–UCC), 08500 Vic, Spain; 4Faculty of Medicine, Universitat de Vic–Universitat Central de Catalunya (UVIC–UCC), 08500 Vic, Spain; 5Clinical and Environmental Infectious Diseases Study Group, Fundació Institut d’Investigació Germans Trias i Pujol, 08916 Badalona, Spain; 6Microbiology Department, Multidisciplinary Inflammation Research Group, Hospital Universitari de Vic, 08500 Vic, Spain; 7Fundació Institut de Recerca i Innovació en Ciències de la Vida i de la Salut de la Catalunya Central, 08500 Vic, Spain; 8Faculty of Health Sciences, Universitat de Vic–Universitat Central de Catalunya (UVIC–UCC), 08500 Vic, Spain

**Keywords:** multiresistant *Enterobacteriaceae*, ESBL-producing *E. coli*, wastewater, contamination of rivers

## Abstract

**Background**: This study aimed to (a) assess the prevalence of multidrug-resistant (MDR) *Enterobacteriaceae* in the waters of two rivers and wastewater treatment plants (WWTPs) in a region of Catalonia, Spain; (b) genetically characterize the MDR strains; and (c) compare extended-spectrum β-lactamase (ESBL)-producing *Escherichia coli* isolates from environmental and human sources. **Methods**: A total of 62 samples were collected from the influent and effluent of 31 WWTPs and 29 river water samples from 11 sites. Simultaneously, 382 hospitalized patients were screened for MDR *Enterobacteriaceae* using rectal swabs. All isolates underwent antibiotic susceptibility testing and whole-genome sequencing. **Results**: MDR *Enterobacteriaceae* were detected in 48.4% of WWTP samples, with 18.5% ESBL-producing *E. coli* and 1.5% (one sample) OXA-48-producing *K. pneumoniae* in influents, and 12.8% ESBL-producing *E. coli* in effluents. In river waters, 5.6% of samples contained ESBL-producing *E. coli* and 1.4% (1 sample) contained VIM-producing *Enterobacter cloacae* complex strains. Among patients, 10.2% (39/382) carried MDR Gram-negative bacilli, of which 66.7% were ESBL-producing *E. coli*. In aquatic ecosystems *E. coli* ST131 (13.3%) and ST162 (13.3%) were the most common strains, while in humans the common were *E. coli* ST131 (33.3%), ST69 (11.1%) and ST410 (7.4%) in humans. The most frequent environmental antibiotic resistance genes (ARG) were *bla*_CTX-M-15_ (24%) and *bla*_TEM-1B_ (20%), while the most common ARGs were *bla*_TEM-1B_ (20.4%), *bla*_CTX-M15_ (18.4%) and *bla*_CTX-M-27_ (14.3%). IncF plasmids were predominant in environmental and human strains. **Conclusions**: ESBL-producing *E. coli* and carbapenemase-producing *Enterobacteriaceae* are present in aquatic environments in the region. Phylogenetic similarities between environmental and clinical strains suggest a possible similar origin. Further studies are necessary to clarify transmission routes and environmental impact.

## 1. Introduction

*Enterobacteriaceae*, particularly *Escherichia coli*, are opportunistic pathogens found as commensals in the intestinal tract of humans and animals and associated with a variety of community- and hospital-acquired infections [1]. Furthermore, these microorganisms can contaminate many different environments, through fecal transmission, subsequently entering water and food consumed by humans [2].

Nowadays, an issue of increasing concern to public health authorities is the spread of multiresistant extended-spectrum β-lactamase (ESBL)- and carbapenemase-producing *Enterobacteriaceae* in the environment, mainly through aquatic ecosystems [3]. Aquatic habitats enable the convergence of bacteria from human, animal and environmental sources, promoting the horizontal transfer of antibiotic-resistance genes (ARGs) and mobile genetic elements [4].

Current wastewater treatment plants (WWTPs) are recognized as a hotspot for the spread of antimicrobial resistance in the environment due to the fact that they receive a complex mixture of municipal, hospital and industrial waste and are not designed to fully remove micropollutants such as antibiotics, antibiotic-resistant bacteria (ARB) and ARGs [5]. Likewise, in the effluents from WWTPs, the biological treatment process creates an environment potentially suitable for resistance development and spread because bacteria are continuously mixed with antibiotics at sub-inhibitory concentrations and simultaneously have a rich supply of nutrients and close cell-to-cell interaction, enhancing the horizontal transfer of ARGs [6].

As a result, these biological components can easily end up discharged into rivers, which then become efficient vehicles for spreading drug-resistant microorganisms over long distances [7,8].

In addition, in WWTP settings, ESBL-producing *Enterobacteriaceae* can be spread through bioaerosols, potentially exposing plant workers and the inhabitants of surrounding areas to harmful microorganisms in the air [2].

Previous studies have revealed the presence of ESBL- and carbapenemase-producing *Enterobacteriaceae* in samples collected from rivers, WWTP effluents and hospital sewage systems. However, there are few epidemiological studies on this issue in the Spanish context [9,10,11,12,13].

Given that Osona, a county in the Catalan province of Barcelona in northeastern Spain, is a region with a high density of pig fattening farms [14], it was conjectured that it might have a high percentage of multiresistant *Enterobacteriaceae* in its aquatic ecosystems.

The aims of our study were (a) to analyze the prevalence of ESBL- and carbapenemase-producing *Enterobacteriaceae* in water samples from the river Ter, the main river flowing through the county, and its principal tributaries, the rivers Gurri and Méder, as well as samples from all of the WWTPs in the county; (b) to genetically characterize any ESBL- and carbapenemase-producing *Enterobacteriaceae* strains detected; and (c) to compare ESBL-producing *E. coli* strains in the natural environment with those detected in local human communities to determine whether they shared a common origin.

## 2. Results

### 2.1. Prevalence Rates

Multiresistant *Enterobacteriaceae* were detected at 15 out of 31 (48.4%) WWTPs (see map in Figure 1) and in 5 out of 29 (17.2%) samples collected from rivers.

Of the 382 human participants screened, 39 (10.2%) carried strains of multiresistant *Enterobacteriaceae* and gram-negative bacilli, of which 26 (66.7%) were ESBL-producing *E. coli*.

### 2.2. Species/Strain Distribution

#### 2.2.1. From WWTP Influent

Of the 64 strains isolated, 12 (18.5%) were identified as ESBL-producing *Escherichia coli* and one strain (1.5%) each of OXA-48-producing *Klebsiella pneumoniae*, SHV-1-producing *K. oxytoca* and ESBL-producing *K. pneumoniae* were also identified.

Other *Enterobacteriaceae* and gram-negative bacilli identified were *A.sobria* (21.6%; 14/64); *A. hydrophila* (13.8%; 9/64); *A. media* (7.7%; 5/64); *P. fluorescens* (6.2%; 4/64); *A. salmonicida* (3.1%; 2/64); *A. veronii* (3.1%; 2/64); *C. freundii* (3.1%;2/64); *P. aeruginosa* (3.1%; 2/64); *P. alcaligenes* (3.1%; 2/64); *P. putida* (3.1%; 2/64); and *A*. spp., *C. koseri*, *O. anthropi*, *P. stutzeri*, *V. parahaemolyticus* (1.5%; 1/64, each).

#### 2.2.2. From WWTP Effluent

Of the 47 strains isolated, six (12.8%) were identified as ESBL-producing *E. coli*, of which one was not related to strains in the influent.

Other Enterobacteriaceae and gram-negative bacilli identified were *A. sobria* (16.7%; 8/47); *A. hydrophila* (12.8%; 6/47); *A. media* (8.5%; 4/47); *P. fluorescens* (6.4%; 3/47); *A. Salmonicida*, *A. veronii*, *A.* spp., *C. freundii*, *P. aeruginosa*, *P. alcaligenes*, *P. stutzeri*, *S. maltophilia* (4.3%; 2/47 each); and *C. koseri*, *Pantoea* spp., *V. parahaemolyticus*, *A. caviae* (2.1%; 1/47 each).

#### 2.2.3. From River Water

Of the 65 strains isolated, four strains (5.6%) of ESBL-producing *E. coli* and 1 strain (1.4%) of VIM-producing *E. cloacae* complex were identified.

Other *Enterobacteriaceae* and gram-negative bacilli identified were *P. fluorescens* (17.0%; 12/65), *A. sobria* (15.5%; 11/65), *P. alcaligenes* (12.7%; 9/65); *P. putida* (12.7%; 9/65), *A. hydrophila* (11.3%; 7/65), *P. aeruginosa* (5.6%; 4/65); *S. fonticola* (2.8%; 2/65); *P. mendocina* (2.8%; 2/65); and *A. salmonicida*, *S. paucimobilis*, *S. maltophilia*, *A. iwoffii* (1.4%; 1/65 each). (See Figure 2, which lists the *Enterobacteriaceae* detected in influent and effluent from WWTPs as well as rivers, and see Appendix A for the genome sequencing of several strains).

#### 2.2.4. From the General Pupulation

Of the 10.2% strains of multiresistant *Enterobacteriaceae* and gram-negative bacilli detected in humans, 26/39 (66.7%) were ESBL-producing *E. coli*, followed by ESBL-producing *K. pneumoniae* (5/39; 12.8%), ESBL-producing *Proteus mirabilis* (2/39; 5.1%), AmpC-producing *C. freundii* (2/39; 5.1%) and to a lesser extent AmpC-producing *E. coli*, AmpC-producing *P. aeruginosa*, multiresistant *P. luteola* and multiresistant *Acinetobacter haemolyticus* (1 each, together representing 10.3%).

### 2.3. Genomic/Phylogenetic Analysis of Multiresistant Enterobacteriaceae

#### 2.3.1. From WWTPs and Rivers

Two strains of carbapenemase-producing *Enterobacteriaceae* were detected, one (an OXA-48-producing *K. pneumoniae*) in the influent of a WWTP and the other (a VIM-producing *E. cloacae* complex) in the river Ter (see Table 1, which shows carbapenemase-producing *Enterobacteriaceae* strains with their respective ST, ARGs and plasmidon replicons).

A strain of ESBL-producing *K. pneumoniae* ST35 was isolated with the following ARGs: *bla*_CTX-M-15_, *bla*_OXA-1_, *bla*_SHV-33_, *aac(6′)-Ib-cr*, *OqxB*, *sul2*, *fosA6*, *tet(A)* and *catB3*.

Thirteen different STs were identified among the 15 isolates of ESBL-producing *E. coli* (see Figure 3, which shows ESBL-producing *E. coli* strains with their respective ST, ARGs, plasmidon replicons and virulence factors) with ST131 and ST162 detected in two isolates (13.3%, each ST), and ST10, ST23, ST86, ST93, ST95, ST361, ST405, ST443, ST2509 (river), ST3052 and ST4353 (river) each detected in only one isolate.

An analysis of the ARG content of the isolates showed that the β-lactam resistance genes *bla*_CTX-M-15_ (24%; 6/25) and *bla*_TEM-1B_ (20%; 5/25) were the most prevalent, followed by *bla*_CTX-M-14_ and *bla*_CTX-M-1_ (2/25; 8% each) detected in two isolates. *Bla*_CTX-M-27_, *bla*_CTX-M-32_, *bla*_CTX-M-42_, *bla*_CTX-M-55_, *bla*_SHV-12_, *bla*_TEM-30_, *bla*_TEM-141_, *bla*_TEM-206_, *bla*_TEM-207_ and *bla*_OXA-1_ genes were detected in only one isolate (1/25; 4% each).

Ciprofloxacin resistance was related to the presence of *qnrS1* (66.6%; 4/6), followed by *aac(6′)-Ib-cr* and *qepA4* (16.7%; 1/6), and aminoglycoside resistance was related to *aac(3)-IV* (40.0%; 2/5), followed by *aac(3)-IId*, *aac(3)-IIa* plus *aac(6′)-Ib-cr* and *aac(3)-IIa* (20.0%; 1/5).

Trimethoprim/sulfamethoxazole resistance was mainly associated with the combinations of *sul* (57.1%; 8/14) and *drfA* genes (42.9%; 6/14). *Sul2* alone was present in two isolates and *sul1* plus *sul2* in one. In addition, *dfrA14* and *dfrA17* were detected in two isolates and *dfrA1* and *dfrA12* in one. Furthermore, seven strains had the *tet(A)* gene and two had the *tet(B)* gene.

Plasmid Finder analysis confirmed 22 different replicons, of which the most common were IncFIB (AP001918) (27.3%; 6/22), IncFII (22.7%; 5/22), IncI1-I(Alpha) (18.2%; 4/22), IncFIA (13.6%; 3/22) and IncFIB (13.6%; 3/22). Most of the isolates contained more than one replicon; one isolate harboured five, while five isolates had three, three isolates had four and two isolates had three replicons. In one isolate, one replicon was found it.

Virulence factor (VF) analysis showed the presence of the following VFs among the isolates: 0–10 VFs (6.6%; 1/15); 11–25 VFs (46.7%; 7/15); 26–40 (46.7%; 7/15).

#### 2.3.2. From the General Population

Regarding ESBL-producing *E. coli* and AmpC-producing *E. coli* (see Appendix A), 16 different STs were identified among the 27 isolates, with ST131 (33.3%; 9/27), ST69 (11.1%; 3/27) and ST410 (7.4%; 2/27) being the most common, followed by ST10, ST12, ST101, ST155, ST162, ST224, ST457, ST609, ST744, ST1193, ST2179, ST3205 and ST12150 (3.7%; 1/27).

An analysis of the ARG content of the isolates showed that the β-lactam resistance genes *bla*_TEM-1B_ (20.4%; 10/49), *bla*_CTX-M15_ (18.4%; 9/49) and *bla*_CTX-M-27_ (14.3%; 7/49) were the most prevalent, followed by *bla*_CTX-M-65_ (6.1%; 3/49), *bla*_OXA-1_ (6.1%; 3/49), *bla*_CTX-M-14_ (4.1%; 2/49), *bla*_CTX-M-32_ (4.1%; 2/49), *bla*_TEM-1A_ (4.1%; 2/49), *bla*_OXA-10_ (4.1%; 2/49), and *bla*_CTX-M-1_, *bla*_SHV-12_, *bla*_CMY-2_, *bla*_DHA-1_, *bla*_TEM-30_, *bla*_TEM-1C_, bla_TEM-126_, *bla*_TEM-186_ and *bla*_TEM-207_ (2.03%; 1/49 each).

Ciprofloxacin resistance was related to the presence of *qnrS1* (50.0%; 5/10) and *aac(6′)-Ib-cr* (30.0% 3/10), followed by *qnrS2* and *qnrB4* (10.0%, 1/10) and aminoglycoside resistance was related to *aac(6′)-Ib-cr* (42.8%; 3/7), followed by *aac(3)-IV*, *ant (2′’)-Ia*, *aac(3)-IId*, *aac(3)-IIa* and *aac(3)-IIa* (14.2%; 1/7 each).

Trimethoprim/sulfamethoxazole resistance was mainly associated with the combinations of *drfA* (45.2%; 19/42) and *sul* (54.8%; 23/42) genes. *Sul2* alone was present in four isolates and *sul1* plus *sul2* in five. In addition, *dfrA17* (42.1%; 8/19), *dfrA1*(21.0; 4/19) and *dfrA14* (15.7%; 3/19) were the most frequent combinations, followed by *dfrA5*, *dfrA7*, *dfrA12* and *dfrA36* (5.2%; 1/19 each).

Plasmid Finder analysis confirmed 35 different replicons, the most common being IncFIB(AP001918) (62.9%; 22/35), Col156 (34.3%; 12/35), IncFIA(31.4%;11/35), IncFII(pRSB107) (25.7%; 9/35), IncFIC(FII) (22.9%; 8/35) and Col (BS512) (14.3%; 5/35).

VF analysis of the isolates yielded the following results: 0–10 VFs (*n* = 0); 11–25 VFs (22.2%; 6/27); 26–40 (77.8%; 21/27).

### 2.4. Comparison of Environmental and Human ESBL-Producing E. coli Strains

A comparison of the phenotype of antibiotic resistance in environmental and human community ESBL-producing *E. coli* strains is illustrated in Figure 4.

A phylogenetic tree was used to assess the genetic relationship between ESBL-producing *E. coli* strains (*n* = 42) from environmental and human community origins (see Figure 5).

All samples identified as ST131 and the sample identified as ST12150x clustered together with a common ancestor, although they were grouped into three distinct subgroups. The evolutionary relationships and degree of Single Nucleotide Polymorphism (SNP) variation among this strains cluster (*n* = 12) were further analyzed (see Table 2) and, using the core SNP genome generated, we created a phylogenetic reconstruction (Figure 6). The topology of the tree reflects a high degree of similarity with high bootstrap statistical support among WE and WI6 environmental samples and patient samples P26 and P22.

The largest subgroup of ST131 strains consisted of those obtained from patients 2, 5, 8, 17, 22 and 24, along with WWTP influent sample 6. In the analysis conducted, differences between 79 and 513 SNPS were detected, with the strain found in patient 2 being the most evolutionarily distant from the other representatives of this cluster (423–513 SNPs).

The second subgroup comprised of strains from patients 3, 4 and 7, with differences between 84 and 113 SNPs. The final subgroup included strains from patient 26 and the WWTP effluent (798 SNPs). This patient resided 17 km from the WWTP in question.

The subgroup comprising WWTP influent 6 and strains from patients 2, 5, 8, 17, 22 and 24 showed a close evolutionary relationship to the subgroup consisting of the strains from patient 26 and the WWTP effluent.

Other ST types were detected in more than one sample. Strains from patient 9 and WWTP influents 1 and 9 shared the same ST (ST162), although slight differences were observed, with patient 9 and WWTP influent 1 being the closest evolutionary match. A similar pattern was observed in the samples isolated from patients 10 and 27, both of which exhibited ST410.

An interesting case was observed with the cluster made up of WWTP influent 11 and patients 6 and 12. WWTP 11 presented a ST10, patient 6 a ST744 and patient 12 a ST10. Although the strains from patient 12 and WWTP influent 11 shared the same ST, the ST in the sample from patient 6 was closer to the ST in the sample isolated from WWTP 11.

## 3. Discussion

Aquatic ecosystems have been recognized as an important medium for the spreading of multiresistant microorganisms, because the discharge of wastewater from human and animal sources into the environment can lead to the bacterial contamination of water bodies. If that water and/or sewage sludge is used for agriculture, the soil and then food products grown in it also become contaminated, facilitating the spread of these dangerous microorganisms among human populations [15].

In this study, we analyzed 62 wastewater samples, including influent and effluent, from 31 WWTPs located throughout the county of Osona in Catalonia. Home to some 164,000 inhabitants, the county is characterized by a high density of pig, goat and sheep farms [16]. Our analysis detected multiresistant *Enterobacteriaceae* in almost half of these WWTPs.

The number of WWTPs tested was significantly higher than in previous Spanish studies (one in [9]; two in [10]; two in [11]; 21 in [4]) and among the species identified, ESBL-producing *E. coli* were the most significant. The fact that we found 12 strains in wastewater influent and 6 in wastewater effluent means the 50% of the strains remained after the wastewater was treated. This suggests that a significant proportion of multiresistant organisms passing through WWTPs will ultimately end up in rivers and agriculture.

ESBL-producing *E. coli* clones ST131 and ST162 were the most prevalent. The high risk ST131 clone in particular is well known to cause both community-and healthcare-associated infections worldwide [17,18]. No overall data are available about the proportions of ST131 isolates (with or without ESBL production) in *E. coli* populations released into the environment, however; it has been detected in WWTP effluent [19] and both influent and effluent in the Czech Republic and also in Switzerland [12] and Japan [17]. ESBL-producing *E. coli* ST162 was also detected recently in wastewater in the Czech Republic [19,20].

Another important finding was the isolation of a strain of OXA-48-producing *K. pneumoniae* ST101 in the influent of a WWTP. To the best of our knowledge, this is the first report of this strain at a WWTP in Spain, though it has been detected in other countries such as Finland and Romania [21,22]. Although the proportion of this strain in the wastewater analyzed in our study was very low, its presence is worrying because this documents the first step of these genes out of the clinical setting into the environment and potentially from there back into the human community [23].

The detection of VIM-producing *E. cloacae* ST45 in a sample taken from the river Ter was of particular interest because as far as we know this is the first report of this strain isolated in river waters in Spain. It should be noted that the location where the river water was sampled was close to a slaughterhouse as well as a sizable flock of ducks, which suggest a possible animal reservoir for the microorganism. This finding differs from those reported in [24], where carbapenemase-producing *Enterobacteriaceae* were recovered from sediment samples, as were two strains of *E. cloacae*, one positive for KPC-2 and the other positive for IMI-2.

This study differed somewhat in its methodology from other studies carried out in the Spanish context, and our results in terms of ESBL-producing *E. coli* environmental strain counts also differ from other findings. For example, our data showed *bla*_CTX-M-15_ and *bla*_TEM-1B_ to be the most prevalent β-lactam resistance genes, followed to a lesser extent by *bla*_CTX-M-14_ and *bla*_CTX-M-1_, while [4] detected mainly *bla*_CTX-M-14_ and *bla*_CTX-M-1_ and [25] detected *bla*_CTX-M-1_. The CTX-M-15 enzyme has been identified in hospitalized and non-hospitalized patients and also in pets, in which CTX-M-15 was the most prevalent CTX-M enzyme [26], as well as in migratory birds [27].

Plasmids of the IncF group were the most prevalent among both environmental and human ESBL-producing *E. coli* isolates. Notably, IncF is the plasmid type most frequently reported in *E. coli* from both human and animal sources, highlighting its central role in the dissemination of antimicrobial resistance within species [28].

Concerning human strains in the general population, out of the 382 participants included in our study, 39 (10.2%) showed evidence of rectal carriage of multiresistant *Enterobacteriaceae*, of which 66.7% were ESBL-producing *E. coli*, with the most prevalent antibiotic-resistant genes being mainly ST131, *bla*_TEM-1B_, *bla*_CTX-M15_ and *bla*_CTX-M-27_. Moreover, these first two genes (*bla*_TEM-1B_ and *bla*_CTX-M15_) are in line with the most prevalent ARGs we found in environmental strains. Carbapenemase-producing Enterobacteriaceae were not detected. To the best of our knowledge, this is the first study showing such a level of prevalence in rural community settings. In the meta-analysis reported in [29] the pooled prevalence of ESBL- producing *E. coli* intestinal carriage in Europe was lower (6%) than in all other continents, but a subgroup meta-analysis carried out separately for every three-year period of the study showed an increasing trend, with a much higher carriage rate in recent years.

Regarding the evolutionary relationship of human and environmental ESBL-producing *E. coli* strains, genome analysis showed samples belonging to the cluster WE-P26 or WI and P22 (but also P5, P24, P17, P8 and P2) to be highly similar, though it remains unclear whether they are associated with a recent outbreak.

The present study has several limitations. The main limitation was the fact we were not able to recover two ESBL-producing *E. coli* strains from rivers for genomic sequencing as well as five ESBL-producing *E. coli* strains from effluents. In the latter case, this is because initially the strains were phenotypically similar to the corresponding influent strains and we therefore did not retain them. Furthermore, we could not perform the genomic sequencing of all Enterobacteriaceae isolated. In addition, the results obtained in this study are from single samples taken at each location, so the presence and abundance of the different strains may be affected by the period of the year when sampling was carried out.

## 4. Materials and Methods

### 4.1. Collection of Aquatic Samples

From November 2020 to July 2023, samples were collected from influent wastewater and outgoing treated effluent at all 31 WWTPs in Osona, yielding 62 samples in total. Samples were collected on an approximately monthly basis. In all cases, 2 L of sample were collected in a sterile 2.5 L container, in order to ensure that all samples, both influent and effluent, were of a uniform volume.

Water samples of identical volume were taken from the rivers Ter, Gurri and Mèder between November 2020 and December 2021, also using sterile 2.5 L containers. Sampling was carried out at 11 different locations in sections of their respective courses where these rivers pass through the main municipalities of Osona (coordinates for these locations are listed in Appendix A). At some sites, samples were taken on more than one occasion, to yield a total of 29 samples altogether. In all cases sampling was carried out when river flows were normal (i.e., not during or after heavy rain) in order to prevent any unusual variation in hydrological conditions from affecting the results [30].

Influent was defined as untreated wastewater from households, hospitals, farms or pharmaceutical industries entering the treatment plants, while effluent was defined as fully treated water exiting the plants and therefore in principle apt for reuse [31].

### 4.2. Microbiological Analysis of Environmental Samples

Once in the microbiology laboratory, the water was filtered using a peristaltic pump and polycarbonate 47 mm and 0.2 μm pore filters. The filter was then placed in a sterile container with 2–3 mL of sterile saline and vortexed. Next, the serum was seeded with a 10 μL loop on selective plates of MacConkey agar, CHROMID^®^ESBL (bioMerieux, Madrid, Spain) and CHROMID^®^CARBA SMART (bioMerieux). Finally, the plates were incubated at 35–37 °C for 24–48 h [32].

The antimicrobial susceptibility of all the enterobacterial isolates recovered in CHROMID^®^ESBL (bioMerieux) and CHROMID^®^CARBA SMART (bioMerieux) was studied by the microdilution method using Vitek2 (BioMerieux^®^, Madrid, Spain). The compounds studied were amoxicillin, piperacillin/tazobactam, cefotaxime, meropenem, aztreonam, gentamicin, amikacin, ciprofloxacin, trimethoprim–sulfamethoxazole, azithromycin, fosfomycin, doxycycline and colistin. Interpretation of the results was performed according to the EUCAST clinical breakpoints when available [32].

Extended-spectrum beta-lactamase-and carbapenemase-producing *Enterobacteriaceae* strains were frozen at −20 °C and then sent to Germans Trias i Pujol Research Institute for whole genome sequencing.

### 4.3. Rectal Carriage of Multiresistant-Enterobacteriaceae Among the General Population

From November 2020 to 2021, during one week in every four months all patients admitted to the University Hospital of Vic, the capital city of Osona, were screened within 24 h of admission for the presence of multiresistant *Enterobacteriaceae* by means of rectal swabs. Screening data from all patients who gave their consent to the use of their results for purposes of this study were included. If the patient was vulnerable or had a condition such as dementia or psychiatric illness, the consent was signed by a close family member of the patient or their legal guardian.

### 4.4. Microbiological Analysis of Rectal Swabs

A rectal swab was collected from each participant with cotton-tipped swabs that were then placed in Stuart swab PS+ Viscose (Deltalab, Rubí, Spain) [33]. Rectal swabs were also plated on selective media, namely MacConkey agar, CHROMID^®^ESBL (bioMerieux) and CHROMID^®^CARBA SMART (bioMerieux) [32].

The antimicrobial susceptibility of all the enterobacterial isolates recovered in CHROMID^®^ESBL (bioMerieux) and CHROMID^®^CARBA SMART (bioMerieux) was studied by the microdilution method using Vitek2 (BioMerieux^®^, Spain). Again, the compounds studied were amoxicillin, piperacillin/tazobactam, cefotaxime, meropenem, aztreonam, gentamicin, amikacin, ciprofloxacin, trimethoprim–sulfamethoxazole, azithromycin, fosfomycin, doxycycline and colistin. Interpretation of the results was again performed according to the EUCAST clinical breakpoints when available [32].

Extended-spectrum betalactamase and carbapenemase-producing Enterobacteriaceae strains were frozen at −20 °C and then sent to Germans Trias i Pujol Research Institute for whole genome sequencing.

### 4.5. DNA Extraction and Quantification

DNA extraction of all isolated bacterial strains was performed using the QIAamp DNA Blood Mini Kit (QIAGEN, Hilden, Germany) following the manufacturer’s instructions. Extractions were quantified by fluorometry (Quantus, Promega, Madison, WI, USA).

### 4.6. Whole Genome Sequencing (WGS)

Seventy-nine strains of *Enterobacteriaceae* and gram-negative bacilli were analyzed by WGS. DNA extractions were normalized at 0.2 ng/µL for library preparation with the Nextera XT DNA Library Preparation Kit (Illumina, San Diego, CA, USA). After the amplification step, the samples were purified with CleanNGS beads (CleanNA, Waddinxveen, The Netherlands). Quality control of libraries was performed using a 2200 TapeStation System (Agilent, Santa Clara, CA, USA). Libraries were individually quantified by fluorimetry (Quantus, Mascot, Australia), pooled and run on the MiSeq system (at 10 pM final concentration containing 10% PhiX) and a million reads per sample were obtained. Sequencing was performed at the Genomic Core Facility at the Centre de Regulació Genòmica in Barcelona.

Raw sequences were imported as paired-end sequences into the KBase platform [34]. Reads were de novo assembled using SPAdes assembler v3.13.0, with default parameters, to obtain contigs as FASTA files. These contigs were used to genotype isolates using Multi Locus Sequence Typing (MLST) [35] and other tools available from the Center for Genomic Epidemiology (Technical University of Denmark), such as ResFinder version 4.7.2, FimTyper version 1.0, PlasmidFinder version 2.1 and VirulenceFinder version 2.0.

The phylogenetic tree was built using the Realphy online tool from whole genome sequence data from all *Escherichia coli*, whether obtained from clinical or environmental sources. The *Escherichia coli* strain K-12 substrain MG1655 was used as a reference to build the tree and was obtained from the National Center for Biotechnology Information [36].

To elucidate the evolutionary relationships and degree of SNP variation among strains from the ST131 cluster we further analyzed them using a SNP core alignment approach. By applying snippy [37] and freebayes [38], we created a SNP core phylogeny and identified the number of SNP differences among the ST131 samples, using *E. coli* K12 as a reference.

In order to determine whether the strains belonged to the same outbreak or not, it was necessary to establish a cutoff regarding genetic divergence (number of SNPs or allele divergence). In accordance with the Sociedad Española de Enfermedades Infecciosas y Microbiología Clínica guidelines for the application of next generation sequencing to pathogenic genomic surveillance [39], we found a threshold of 10 SNPs to determine the same outbreak origin for the *E. coli* strains.

Basing ourselves on the core SNP genome generated, we were able to create a phylogenetic reconstruction using appropriate evolutionary models and bootstrapping parameters using the IQ-Tree software (v2.3.6) [40].

## 5. Conclusions

Our study demonstrates the presence of ESBL- and carbapenemase-producing *Enterobacteriaceae* in water from the river Ter, which flows through an area of Catalonia, Spain, that is characterized by intensive animal husbandry. Noteworthy findings are the first detection of a VIM-producing *E. cloacae* ST45 strain and the first detection of an OXA-48-producing *K. pneumoniae* strain in WWTP influent in this region. Also significant is the fact that we isolated ESBL-producing *E. coli* in almost half of all the WWTPs in the county.

Likewise, the presence of multiresistant *Enterobacteriaceae* in WWTP effluent as revealed here points to insufficient clearance during treatment processes. These findings underscore the need for improved wastewater management to prevent the release of resistant bacteria into natural water bodies and thereby reduce the risk of subsequent human infection.

Finally, several environmental and human strains of the ESBL-producing *E. coli* identified in this study were shown to have a common ancestor, which suggests that the resistance may have originated in one environment and then spread to the other.

Therefore, further studies which involve long-term and seasonal sampling in rivers along with the determination of antibiotics concentrations in aquatic ecosystems are necessary in order to draw more robust conclusions about the persistence and spread of these multiresistant *Enterobacteriaceae*.

## Figures and Tables

**Figure 1 antibiotics-14-00753-f001:**
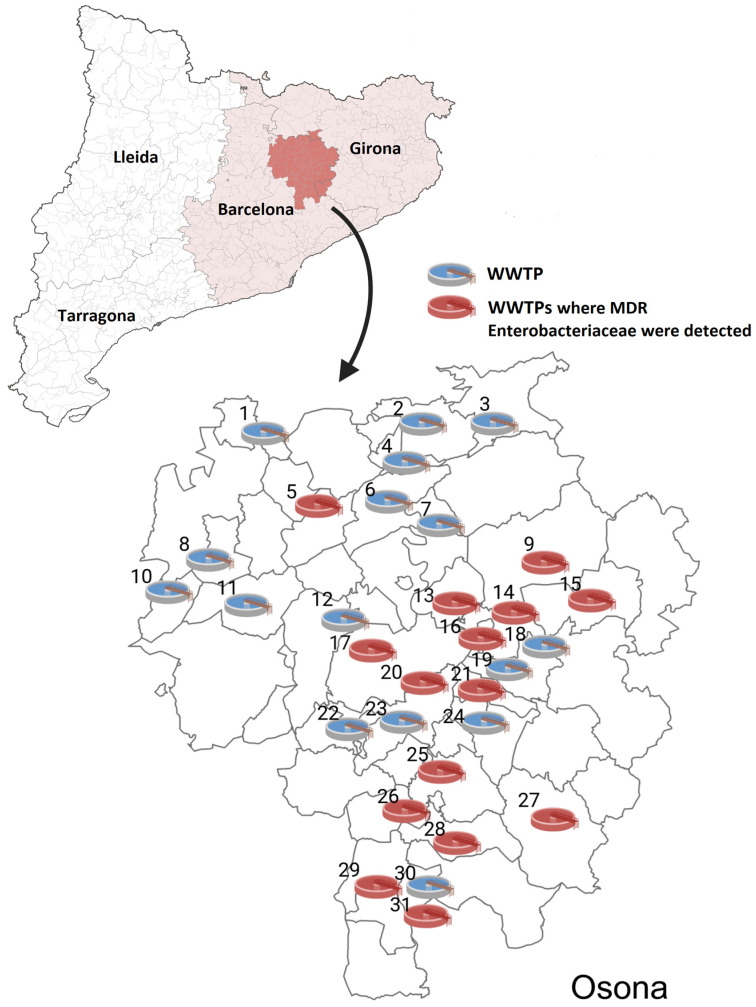
Upper map shows the location of the county of Osona in Catalonia, Spain. Lower map shows the locations within Osona of WWTPs where samples were taken, with those where multiresistant Enterobacteriaceae were detected coloured red. 1: WWTP Alpens; 2: WWTP Santa Maria de Besora; 3: WWTP Vidrà; 4: WWTP Sant Quirze de Besora; 5: WWTP Sant Boi del Lluçanès; 6: WWTP Can Branques; 7: WWTP Vall del Ges—Torelló; 8: WWTP La Blava; 9: WWTP Santa Maria de Corcó—L’Esquirol; 10: WWTP Prats de Lluçanès; 11: WWTP Olost; 12: WWTP Bingrau; 13: WWTP Manlleu; 14: WWTP Les Cases Noves (Les Masies de Roda); 15: WWTP Tavertet; 16: WWTP Roda de Ter; 17: WWTP Serrabonica—Gurb; 18: WWTP Fussimanya; 19: WWTP Tavèrnoles; 20: WWTP Vic; 21: WWTP Folgueroles; 22: WWTP Santa Eulàlia de Riuprimer; 23: WWTP Sentfores—La Guixa; 24: WWTP Vilalleons; 25: WWTP Taradell; 26: WWTP Tona; 27: WWTP Viladrau; 28: WWTP Seva; 29: WWTP Centelles; 30: WWTP Muntanyà (Seva); 31: WWTP Masia Perafita.

**Figure 2 antibiotics-14-00753-f002:**
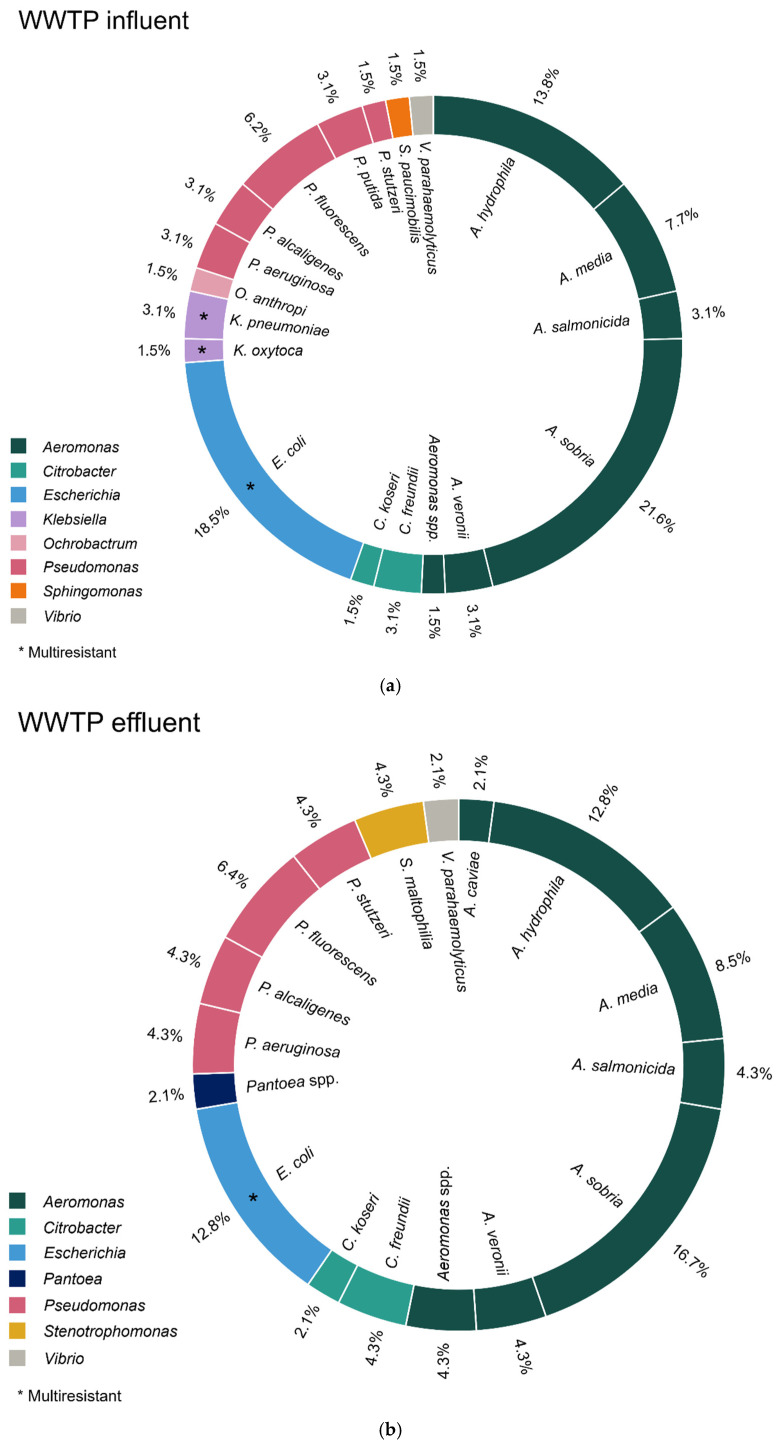
Species of *Enterobacteriaceae* and gram-negative bacilli family isolated from WWTP influent (**a**) and effluent (**b**) and river water (**c**) samples. (**a**) Number of isolates (*n*): *A. sobria* (*n* = 14); ESBL-producing *E. coli* (*n* = 12); *A. hydrophila* (*n* = 9); *A. media* (*n* = 5); *P. fluorescens* (*n* = 4); ESBL- and OXA-48-producing *K. pneumoniae* (*n* = 2); *A. salmonicida* (*n* = 2); *A. veronii* (*n* = 2); *C. freundii* (*n* = 2); *P. aeruginosa* (*n* = 2); *P. alcaligenes* (*n* = 2); *P. putida* (*n* = 2), *A.* spp., *C. koseri*, *O. anthropi*, *P. stutzeri*, *V. parahaemolyticus* (*n* = 1, each). (**b**) Number of isolates (*n*): *A. sobria* (*n* = 8); ESBL- producing *E. coli* (*n* = 6); *A. hydrophila* (*n* = 6); *A. media* (*n* = 4); *P. fluorescens* (*n* = 3); *A. salmonicida*; *A. veronii*; *A.* spp.; *C. freundii*; *P. aeruginosa*; *P. alcaligenes*; *P. stutzeri*; *S. maltophilia* (*n* = 2, each); *C. koseri*; *Pantoea* spp.; *V. parahaemolyticus*, *A. caviae* (*n* = 1, each). (**c**) Number of isolates (*n*): *P. fluorescens* (*n* = 12); *A. sobria* (*n* = 11); *P. alcaligenes* (*n* = 9); *P. putida* (*n* = 9); *A. hydrophila* (*n* = 8); *A. caviae* (*n* = 5); ESBL-producing *E. coli* (*n* = 4); *P. aeruginosa* (*n* = 4); *S. fonticola* (*n* = 2); *P. mendocina* (*n* = 2); VIM-producing *E. cloacae* complex; *A. salmonicida*, *S. paucimobilis*, *S. maltophilia*, *A. iwoffii* (*n* = 1, each).

**Figure 3 antibiotics-14-00753-f003:**
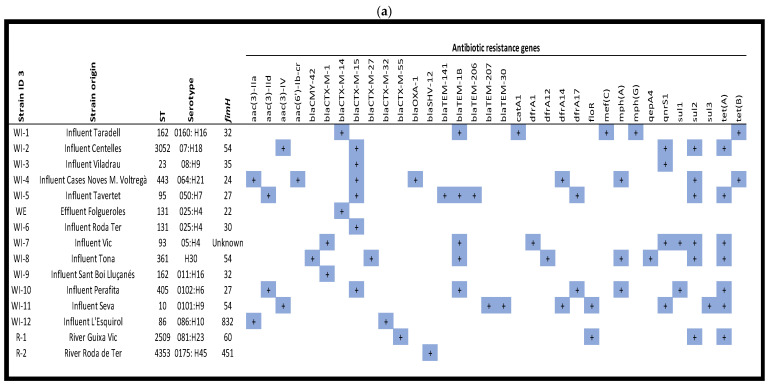
ESBL-producing *E. coli* strains with their respective origin, ST, antibiotic resistance genes (**a**), plasmid replicons (**b**) and virulence factors (**c**).

**Figure 4 antibiotics-14-00753-f004:**
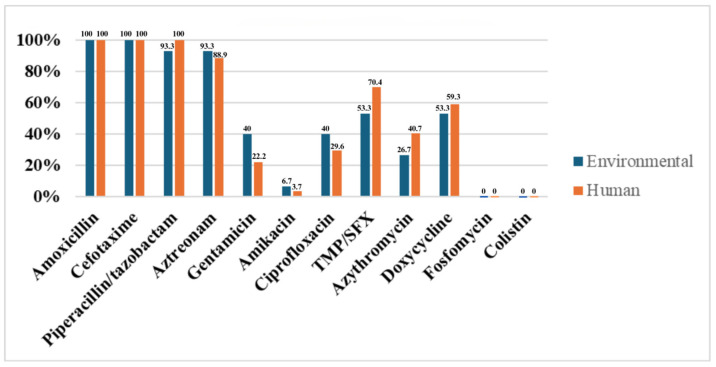
Phenotypic analysis of antibiotic resistance in environmental and human community ESBL-producing *E. coli* strains.

**Figure 5 antibiotics-14-00753-f005:**
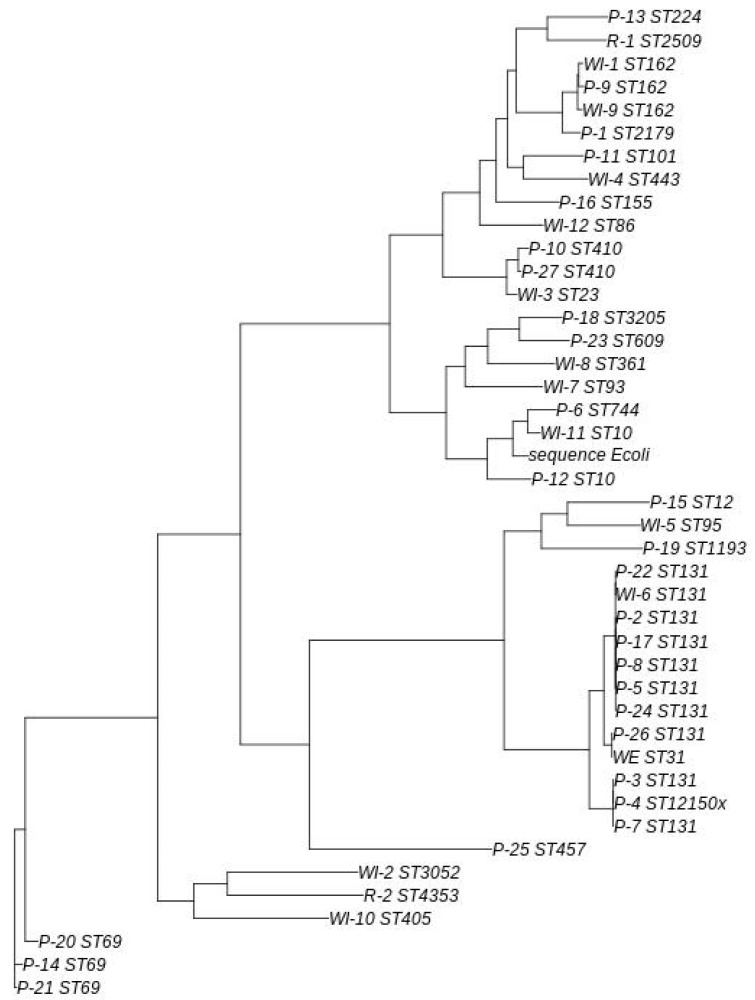
Phylogenetic tree of environmental and human community strains of ESBL-producing *E. coli*.

**Figure 6 antibiotics-14-00753-f006:**
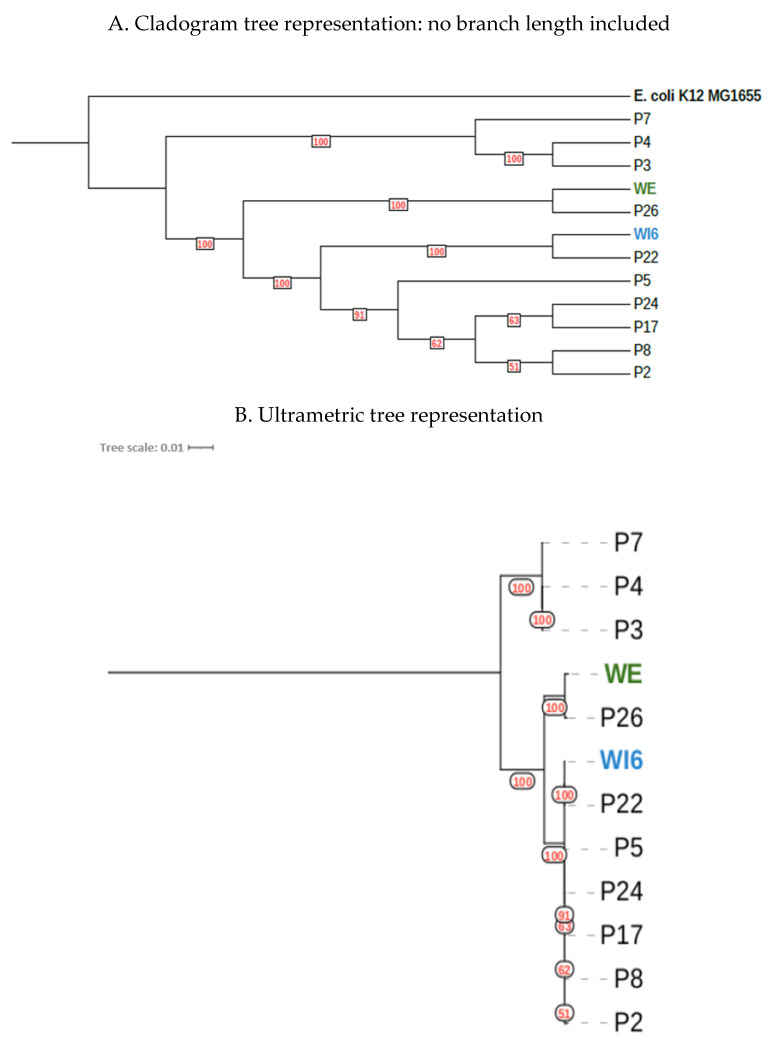
SNP Phylogenetic tree of the samples analyzed using *E. coli* K12 as a reference. In red, bootstrap percentage support. In blue and green, samples of interest. (**A**) Cladogram tree with no genetic distance reflected in the branch length. (**B**) Ultrametric tree with genetic distance branch length correction.

**Table 1 antibiotics-14-00753-t001:** Carbapenemase-producing *Enterobacteriaceae* detected in samples taken from WWTPs and rivers, with their respective origin, ST, antibiotic resistance genes and plasmid replicons.

Sample Site	Species	SpaType	ARG(s) Present	Plasmids Replicons
WWTP influent	OXA-48-producing *K. pneumoniae*	101	*bla*_OXA-48_, *bla*_OXA-1_, *bla*_TEM-1B_, *bla*_SHV-106_, *bla*_CTX-M-15_, *aac(6′)-Ib-cr*, *aac(3)-IId*, *sul2*, *OqxB*, *OqxA*, *dfrA14*, *tet(D)* and *fosA*	Col440I, ColpVC, IncFIB(K), IncFII(pKP91), IncL and IncR
River	VIM-producing *E. cloacae* complex	45	*bla*_VIM-1_, *bla*_ACT-15_, *bla*_TEM-1B_, *bla*_OXA-1_, *bla*_CTX-M-9_, *ant (2″)-Ia*, *aac(6′)-Ib3*, *aac(6′)-Ib-cr*, *qnrB5*, *qnrA1*, *qnrB81*, *qnrB19*, *sul1*, *sul2*, *dfrA14*, *dfrB1*, *tet(A)*, *catB3* and *mcr-9*	Col(pHAD28), IncFIB(pECLA), IncFII(pECLA), IncHI2, and IncHI2A

**Table 2 antibiotics-14-00753-t002:** SNP pairwise distance between samples.

	P17	P22	P24	P26	P2	P3	P4	P5	P7	P8	Ref.	WE	WI6
**P17**	0	-	-	-	-	-	-	-	-	-	-	-	-
**P22**	179	0	-	-	-	-	-	-	-	-	-	-	-
**P24**	86	165	0	-	-	-	-	-	-	-	-	-	-
**P26**	4651	4680	4661	0	-	-	-	-	-	-	-	-	-
**P2**	440	513	452	4708	0	-	-	-	-	-	-	-	1
**P3**	10,520	10,530	10,520	10,630	10,700	0	-	-	-	-	-	-	-
**P4**	10,510	10,520	10,520	10,620	10,700	84	0	-	-	-	-	-	-
**P5**	89	148	89	4638	423	10,480	10,490	0	-	-	-	-	-
**P7**	10,500	10,510	10,490	10,610	10,690	113	101	10,470	0	-	-	-	-
**P8**	96	159	108	4661	430	10,510	10,500	79	10,490	0	-	-	-
**Ref.**	95,810	95,850	95,800	96,980	95,620	95,620	95,630	95,810	95,600	95,820	0	-	-
**WE**	4794	4779	4782	798	4820	10,770	10,770	4761	10,760	4766	97,120	0	-
**WI6**	141	146	149	4660	503	10,510	10,510	134	10,500	147	95,810	4779	0

## Data Availability

The data presented in the present study are available on request from the corresponding author.

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
