# Peer review of "Environmental Dispersion of Multiresistant Enterobacteriaceae in Aquatic Ecosystems in an Area of Spain with a High Density of Pig Farming"

_antibiotics, 2025, doi:10.3390/antibiotics14080753_

Round 1
Reviewer 1 Report
Comments and Suggestions for Authors
Following requests are necessary to be addressed to improve quality of the study.
Title:
- Is the isolated Enterobacteriaceae resistant to multiple classes of antibiotics, or only to extended-spectrum β-lactamases (ESBLs)? Additionally, what is the rationale for focusing on this particular class of antibiotics?
- Azithromycin and doxycycline are widely used in the treatment of human gastrointestinal infections caused by Enterobacteriaceae. Why were these antibiotics not included in the investigation?
Abstract:
- Line 28, In your opinion, are there any other pathways for the transmission of antibiotic resistance besides aquatic ecosystems?
- Line 30, "Could you clarify why your study did not include samples from pig, sheep, or goat farm environments?
- Line 39, there are no results for other types of Enterobacteriaceae rather than E coli
- Line 45, you need here clrear explain reveals of study
- No results related to antibiotic resistant genes
- No explain how isolated strain is related to human strain
- In this section you need to explain the limit of the study and suggest further studies
Introduction:
- In this section, you need to explain hotspot area for dispersion of antibiotic resistance
- In this section, in this section, you should include a paragraph describing the possible mechanisms by which virulence and antibiotic resistance genes are transferred between bacterial genera, as well as the factors that may accelerate this transmission.
- Results:
- Line 109: is Pseudomonas fluorescens belong to Enterobacteriaceae family?
- Line 113, this results need to remove.
- Line 171 , can you clarify why take rectal swab
- There are no detailed results regarding antibiotic resistance genes and their associated mobile genetic elements
- Line 191 , "Trimethoprim/sulfamethoxazole is not classified as an ESBL-targeted antibiotic, and this study did not aim to investigate its occurrence in the environment, as reflected in the study's title and objectives
- Overall, the results in this section need to be reorganized to improve clarity and better reflect the aims of the study
Discussion
- In this section, you need to provide clear statements and appropriate explanations for the following points
- The more common antibacterial that use to treat infection with Enterobacteriaceae in human and animal in the selected area
- why you selected ESBL
- Which type of genetic element is associated with the selected antibiotic resistance gene?
- Clarify the potential impact of environmental Enterobacteriaceae on human
Methods:
- Line 298 , It is necessary to provide more details regarding the sampling process.
- Line 341 , Please specify the number of samples analyzed by whole genome sequencing
Conclusions
- This section should clearly state the study’s aim and emphasize its potential impact

Author Response
Reviewer 1
Title:
- Is the isolated Enterobacteriaceae resistant to multiple classes of antibiotics, or only to extended-spectrum β-lactamases (ESBLs)? Additionally, what is the rationale for focusing on this particular class of antibiotics?
Answer: We appreciate your questions. We wanted to know most data possible regarding multiresistant Enterobacteriaceae, for this reason, we investigated and tested ESBL, and the rest of antibiotics usually used in our clinical daily practice.
- Azithromycin and doxycycline are widely used in the treatment of human gastrointestinal infections caused by Enterobacteriaceae. Why were these antibiotics not included in the investigation?
Answer: We have added this profile of resistance although we have to clarify that doxycycline is not an usual antibiotic in the treatment of human gastrointestinal infections caused by Enterobacteriaceae in our area.
Abstract:
- Line 28, in your opinion, are there any other pathways for the transmission of antibiotic resistance besides aquatic ecosystems?
Answer: Thank you for your interest in this topic. In my opinion and in accordance with previous articles (doi.org/10.1016/j.envpol.2023.122476; doi: 10.4081/ijfs.2019.7956), ESBL- and carbapenemase - producing Enterobacteriaceae have emerged in healthy food-producing animals, household pets and food products such as meat, fish, and raw milk, as other sources for the transmission of antibiotic resistance. The aquatic environment as a possible vector of multidrug-resistant bacteria was the exclusive focus of our investigation. Other ecosystems might be the subject of future research.
- Line 30, "Could you clarify why your study did not include samples from pig, sheep, or goat farm environments?
Answer: We started this study focused on the aquatic ecosystems and its possible impact on human health. Currently, we are working on a new project studying multiresistant Enterobacteriaceae in these animals and we will be able to compare all these environmental, animal and humas strains in the future.
- Line 39, there are no results for other types of Enterobacteriaceae rather than E. coli
Answer: We appreciate your comment. The abstract has been updated to incorporate the main results, enhancing the study’s clarity and scientific relevance.
- Line 45, you need here clear explain reveals of study
Answer: Thank you for your valuable comments. As we mention previously, we have revised the abstract to include key results.
- No results related to antibiotic resistant genes
Answer: We have added antibiotic resistant genes following your advice.
- No explain how isolated strain is related to human strain
Answer: Thank you for your comments. We have added to the results the relationship between the strains isolated in aquatic ecosystems and the human strains.
- In this section you need to explain the limit of the study and suggest further studies
Answer: Thank you for your valuable comments. We have added further studies which involve long-term and seasonal sampling in rivers along with the determination of antibiotics concentrations in aquatic ecosystems are necessary in order to improve conclusions about persistence and spread of these multiresistant Enterobacteriaceae.
Introduction:
- In this section, you need to explain hotspot area for dispersion of antibiotic resistance.
Answer: Thank you for your comment, so we have added some new data regarding this.
- In this section, you should include a paragraph describing the possible mechanisms by which virulence and antibiotic resistance genes are transferred between bacterial genera, as well as the factors that may accelerate this transmission.
Answer: We appreciate your suggestion, for this reason, we have included a paragraph explaining this and we have added two new references supporting these data.
Results:
- Line 109: is Pseudomonas fluorescens belong to Enterobacteriaceae family?
Answer: Thank you for your question. The reviewer is right and the genus Pseudomonas, Stenotrophomonas and Sphingomonas do not belong to the enterobacterial family, so we have clarified these data.
- Line 113, this results need to remove.
Answer: We have removed this line following your advice as well as the part of the discussion, where we mentioned our findings related to MRSA.
- Line 171, can you clarify why take rectal swab
Answer: Thank you for your valuable comments. We have modified ‘rectal swabs’ by ‘human population’ that we consider to be more accurate. In accordance with our hospital's protocols, rectal swabs are employed for the detection of multidrug-resistant Enterobacteriaceae carriers; therefore, this method was used in our study.
- There are no detailed results regarding antibiotic resistance genes and their associated mobile genetic elements
Answer: Thank you for your comments. We have tried to explain these genes in results and tables.
- Line 191, "Trimethoprim/sulfamethoxazole is not classified as an ESBL-targeted antibiotic, and this study did not aim to investigate its occurrence in the environment, as reflected in the study's title and objectives
Answer: Thank you for your comment. We have expanded the Results section to include a phenotypic analysis of antibiotic resistance, focusing on the antibiotics most commonly used against Gram-negative bacilli.
- Overall, the results in this section need to be reorganized to improve clarity and better reflect the aims of the study
Answer: Thank you for your comment. We have tried to improve clarity of the results.
Discussion
In this section, you need to provide clear statements and appropriate explanations for the following points:
- The more common antibacterial that use to treat infection with Enterobacteriaceae in human and animal in the selected area
Answer: The antibiotic treatment for human infections by multiresistant Enterobacteriaceae in our area depends on the kind and severity of infection we would use carbapenems and if possible a quinolone or cotrimoxazole we would de-escalate.
- Why you selected ESBL
Answer: We selected ESBL because there are some studies which evaluate a rectal carriage of ESBL-producing Enterobacteriaceae in livestock such as Abreu R, et al. Characterisation of Multiresistant Bacterial Strains Isolated in Pigs from the Island of Tenerife. doi: 10.3390/vetsci9060269., therefore, as we live in a region with a high density of pig fattening farms, we wanted to determine the prevalence of ESBL in aquatic ecosystems and its influence in human population.
- Which type of genetic element is associated with the selected antibiotic resistance gene?
Answer: Thank you for your comments. The most frequent environmental antibiotic resistance genes (ARG) were blaCTX-M-15 (24%) and blaTEM-1B (20%), while the most common ARGs were blaTEM-1B (20.4%), blaCTX-M15 (18.4%) and blaCTX-M-27 (14.3%).
- Clarify the potential impact of environmental Enterobacteriaceae on human
Answer: Thank you for your comments. The potential impact of environmental Enterobacteriaceae on human health has been addressed and incorporated into the corresponding section of the manuscript.
Methods:
- Line 298, It is necessary to provide more details regarding the sampling process.
Answer: We have provided more details regarding the sampling process, following your suggestion.
- Line 341, Please specify the number of samples analyzed by whole genome sequencing
Answer: We analysed 79 strains of Enterobacteriaceae and gran-negative bacilli by whole genome sequencing, so we have added this data accordingly. Furthermore, we analysed the strain of MRSA by WGS, but we have removed this according to your previous suggestion.
Conclusions
- This section should clearly state the study’s aim and emphasize its potential impact
Answer: Thank you for your comments. The potential impact of environmental Enterobacteriaceae on human health has been addressed and incorporated into the corresponding section of the manuscript.
Reviewer 2 Report
Comments and Suggestions for Authors
Peer Review for: „Environmental dispersion of multiresistant Enterobacteriaceae in aquatic ecosystems in an area of Spain with a high density of pig farming“
Overall Evaluation:
This is a well-structured and timely study investigating the prevalence and molecular characteristics of multidrug-resistant Enterobacteriaceae in rivers and wastewater treatment plants (WWTPs) in the Osona region of Catalonia, Spain - an environment known for its high livestock density. The manuscript addresses a critical One Health issue by establishing a link between environmental and human reservoirs of antibiotic resistance and would be a valuable contribution to the literature once the below revisions have been taken into account.
Major Recommendations:
- The study is based on single sampling events per location, which limits the insight into temporal variability. Long-term or seasonal sampling would significantly improve conclusions about persistence and spread. This should be emphasised more clearly in the discussion and proposed as future work.
- The manuscript does not contain quantitative/statistical comparisons (e.g. prevalence differences between influx and efflux, environmental and human isolates). Consider basic statistical tests (e.g., chi-square) to support statements about prevalence or risk patterns.
- The authors report that some isolates were not retained for sequencing, leading to data gaps. Propose future protocols to preserve all isolates, regardless of the initial phenotype match.
- While the ST131 clustering is intriguing, the interpretation could benefit from clarification. Clearly define thresholds for “evolutionary difference” in the phylogenetic tree and indicate the degree of SNP variation.
- The article notes ARG presence on plasmids but does not deeply explore implications of horizontal gene transfer or plasmid epidemiology. Expand the discussion on the role of specific plasmids (e.g., IncFIB, IncFII) in AMR dissemination.
- Ethical Clarification - Clarify if informed consent was obtained for rectal swabs in vulnerable populations (elderly, hospitalized).
Author Response
Reviewer 2
Overall Evaluation:
This is a well-structured and timely study investigating the prevalence and molecular characteristics of multidrug-resistant Enterobacteriaceae in rivers and wastewater treatment plants (WWTPs) in the Osona region of Catalonia, Spain - an environment known for its high livestock density.
The manuscript addresses a critical One Health issue by establishing a link between environmental and human reservoirs of antibiotic resistance and would be a valuable contribution to the literature once the below revisions have been taken into account.
Answer: We appreciate your insight.
Major Recommendations:
- The study is based on single sampling events per location, which limits the insight into temporal variability. Long-term or seasonal sampling would significantly improve conclusions about persistence and spread. This should be emphasised more clearly in the discussion and proposed as future work.
Answer: We are grateful for your contribution, so we have proposed an idea for a future work in the part of conclusion because we will start a new study to determine the antibiotic concentration in river water samples and again the isolation of multiresistant Enterobacteriaceae in every season.
- The manuscript does not contain quantitative/statistical comparisons (e.g. prevalence differences between influx and efflux, environmental and human isolates). Consider basic statistical tests (e.g., chi-square) to support statements about prevalence or risk patterns.
Answer: We appreciate your suggestion. We agree that statistical comparisons can enhance the robustness of prevalence-related conclusions. However, in our case, the sample size was too small to allow meaningful or reliable statistical analysis without compromising validity. Therefore, we chose to present the data in a descriptive manner. We acknowledge this limitation in the revised manuscript and suggest that future studies with larger datasets will be necessary to perform formal statistical comparisons.
- The authors report that some isolates were not retained for sequencing, leading to data gaps. Propose future protocols to preserve all isolates, regardless of the initial phenotype match.
Answer: We appreciate your point of view and advice. We have to explain that most of our team are involved in dealing with patients and it has been our first study related to analysing aquatic samples, for this reason, we had this mistake. Concomitant to this period, we had to treat numerous patients with SARS-COV-2 and analyse multiple samples, so it was not an easy process.
As previously we metioned, we are working on a new study to determine the antibiotic concentration in river water samples and again the isolation of multiresistant Enterobacteriaceae in every season, so we will take into consideration this.
- While the ST131 clustering is intriguing, the interpretation could benefit from clarification. Clearly define thresholds for “evolutionary difference” in the phylogenetic tree and indicate the degree of SNP variation.
Answer: To elucidate the evolutionary relationships and degree of SNP variation among strains from the ST131 cluster we further analyze them using a SNP core alignment approach. By applying snippy, freebayes and IQtree we created a SNP core phylogeny and identified the number of SNP differences among the ST131 samples and E. coli K12, as a reference.
Topology of the tree reflects a high degree of similarity with high boostrap statistic support among WE and WI6 environmental samples and patient samples, P26 and P22, respectively, SNP distance among these particular pairs prevent us from stating that the they have the same outbreak origin (WE/P26 = 798 and WI6/P22=149) reflecting a share genetic background but not a recent transmisión.
We have added all these data to the text with their corresponding table and figure.
- The article notes ARG presence on plasmids but does not deeply explore implications of horizontal gene transfer or plasmid epidemiology. Expand the discussion on the role of specific plasmids (e.g., IncFIB, IncFII) in AMR dissemination.
Answer: We have added a explanation about this following your suggestion.
- Ethical Clarification - Clarify if informed consent was obtained for rectal swabs in vulnerable populations (elderly, hospitalized).
Answer: Thank you for your suggestion. If the patient was vulnerable or had a condition such as dementia or psychiatric illness the consent was signed by a close family member of the patient or their legal guardian. We have added this explanation to the manuscript following your advice.
Reviewer 3 Report
Comments and Suggestions for Authors
Dear Authors, thank you for your efforts. I have included comments and suggestions that I hope will help strengthen your manuscript
Line 39: What does it mean by VIM-producing E. cloacae complex?
Line 84: Results are very difficult to understand. Make it easy to present the sample source, organisms identified, genes, plasmid, ST in a table will be much more easier
Line 298-299: What was the sampling strategy? Was it individual or pooled samples?
Line 300-301: How did you collect water samples? What was the logics to select these 11 points? Was it pooled samples?
Line 310: What is the reference for this microbiological analysis of environmental samples?
Line 311: In a sterile 2-litre container, how much water did you collect?
Line 314-316: How did you select colonies for the antimicrobial susceptibility testing?
Line 317-320: Where is the result of antimicrobial susceptibility testing?
Line 325: Why were rectal swabs collected for one week every four months?
Line 330: What is the reference for this microbiological analysis of rectal samples?
Line 340: For how many samples did you perform whole-genome sequencing (WGS)?
Comments on the Quality of English LanguageThe English could be improved
Author Response
Reviewer 3
Dear Authors, thank you for your efforts. I have included comments and suggestions that I hope will help strengthen your manuscript.
Answer: Thank you for your insight.
Line 39: What does it mean by VIM-producing E. cloacae complex?
Answer: VIM-producing E. cloacae complex means Verona integron-encoded metallo-beta-lactamase - producing E. cloacae complex. Likewise, we had included this abbreviation with its corresponding meaning in the list of abbreviations.
Line 84: Results are very difficult to understand. Make it easy to present the sample source, organisms identified, genes, plasmid, ST in a table will be much more easier
Answer: We appreciate your advice, so we have designed a brief table 1 providing details regarding carbapenemase-producing Enterobacteriaceae, we have removed text about this to make it easier to read this part and we have designed the current table 2 providing details about ESBL-producing E. coli as you mention previously.
Line 298-299: What was the sampling strategy? Was it individual or pooled samples?
Answer: We have provided more details regarding the sampling process. We collected the samples on an approximately monthly basis and at each visit we took 2 litres in a sterile 2.5L container, in order to have all samples of the same size, both influent and effluent.
Line 300-301: How did you collect water samples? What was the logics to select these 11 points? Was it pooled samples?
Answer: Samples were collected directly from the riverbeds of the studied rivers, with sampling sites strategically selected at locations such as river outlets adjacent to villages and in proximity to a pig slaughterhouse.
Line 310: What is the reference for this microbiological analysis of environmental samples?
Answer: Thank you for your comments. We have added this reference European Committee on Antimicrobial Susceptibility Testing (EUCAST). Breakpoint tables for interpretation of MICs and zone diameters. 2016, Version 6.0 EUCAST: Available from: http://www.eucast.org/ clinical–breakpoints/.
Line 311: In a sterile 2-litre container, how much water did you collect?
Answer: We have modified this point and we used a sterile 2.5L and we collected 2 litres of water in all the samples.
Line 314-316: How did you select colonies for the antimicrobial susceptibility testing?
Answer: All morphologically distinct colonies growing on selective media were analyzed. Consequently, multiple Gram-negative bacilli were isolated from several samples. However, the scope of this study was limited to the characterization of ESBL-producing Escherichia coli and carbapenemase-producing Enterobacteriaceae.
Line 317-320: Where is the result of antimicrobial susceptibility testing?
Answer: Thank you for your comments. A figure illustrating the antibiotic resistance profiles of the ESBL-producing E. coli strains isolated has been added to the Results section.
Line 325: Why were rectal swabs collected for one week every four months?
Answer: In coordination with the Epidemiology Department of the University Hospital of Vic, it was determined that conducting one-week prevalence cutoffs was operationally more efficient and did not compromise the integrity of sample analysis.
Line 330: What is the reference for this microbiological analysis of rectal samples?
Answer: We have added these references: European Committee on Antimicrobial Susceptibility Testing (EUCAST). Breakpoint tables for interpretation of MICs and zone diameters. 2016, Version 6.0 EUCAST: Available from: http://www.eucast.org/ clinical–breakpoints/. and Red Nacional de Vigilancia Epidemiológica (RENAVE). Protocolo de vigilancia y control de microorganismos multirresistentes o de especial relevancia clínico-epidemiológica (Protocolo MMR). Madrid, 2016 (https://cne.isciii.es/documents/d/cne/protocolommr_nov2017_rev_abril2019)
Line 340: For how many samples did you perform whole-genome sequencing (WGS)?
Answer: We analysed 79 strains by whole genome sequencing, so we have added this data accordingly.
Reviewer 4 Report
Comments and Suggestions for Authors
This paper has the potential to contribute important findings to the field of environmental AMR surveillance, but the manuscript structure, particularly in the Results section, needs a major overhaul to improve readability and narrative flow. Better organization will help the reader appreciate the scope and implications of the findings.
- The Results section jumps between data sources (WWTP influent, effluent, rivers, human samples) without clear thematic progression or consistent structuring. The sections are overloaded and try to do too much at once—reporting findings from influent, effluent, rivers, and incidental findings like MRSA. It would be better to reorganize the Results into distinct, clearly labelled subsections and structure content as: Detection/prevalence rates, Species/strain distribution, Resistance profiles, Genomic/phylogenetic analysis
- Tables 1 and 2 provide extensive genomic data on coli isolates from environmental and human sources. These tables are comprehensive and technically valuable. However, despite the depth of information, the tables are not fully utilized in the main text. The Results sections that reference them (2.3 and 2.4) tend to reiterate individual data points (e.g., "13 different STs were identified") without drawing meaningful synthesis or interpreting broader trends—such as which STs are shared across environmental and clinical settings, or which plasmid types are most strongly associated with multidrug resistance. Furthermore, the lack of summary statistics in the text (e.g., percentages of isolates carrying major ARGs or replicon families) forces readers to manually scan the tables to extract key insights. This weakens the tables' impact and makes it harder to assess patterns.
- There is a Lack of Integration Between Environmental and Human Finding. Although Section 2.4 covers rectal swab results, it feels disconnected from the environmental findings. Add a transitional paragraph or summary at the end of Section 2 summarizing similarities and suggesting possible links (currently deferred until the Discussion).
- The manuscript does not specify which antibiotics were used to assess multidrug resistance, nor does it report MIC values or susceptibility profiles beyond genetic data. This is a significant omission. The interpretation of multidrug resistance should be supported by phenotypic susceptibility testing, ideally referencing EUCAST/CLSI breakpoints. A brief table or supplementary data listing tested antibiotics and resistance rates would improve transparency.
- While WGS was performed on select isolates, the data remains underutilized and largely descriptive. To fully leverage the potential of this data and improve the impact of the study, additional genomic analyses should be considered. Such as- mobile genetic element analysis to identify integrons, transposons, or insertion sequences that may facilitate the transfer of resistance genes, and plasmid reconstruction to assess the co-localization of multiple resistance determinants on the same plasmid. A pan-genome analysis could reveal gene gain or loss events and functional differences between isolates from environmental and human sources. Furthermore, comparative resistome profiling using clustering or ordination methods (e.g., PCA or heatmaps) would help highlight similarities or divergence across sample types.
- Additionally, qPCR-based quantification of key ARGs could substantially enhance the scientific depth of the manuscript.
- Figure 2 does not indicate the number of isolates (n) per species. The caption should also include these counts.
- All microbial names should be italicized. After first use, only the first letter of the genus should be used.
Author Response
Reviewer 4
This paper has the potential to contribute important findings to the field of environmental AMR surveillance, but the manuscript structure, particularly in the Results section, needs a major overhaul to improve readability and narrative flow. Better organization will help the reader appreciate the scope and implications of the findings.
Answer: We appreciate your comments.
- The Results section jumps between data sources (WWTP influent, effluent, rivers, human samples) without clear thematic progression or consistent structuring. The sections are overloaded and try to do too much at once—reporting findings from influent, effluent, rivers, and incidental findings like MRSA. It would be better to reorganize the Results into distinct, clearly labelled subsections and structure content as: Detection/prevalence rates, Species/strain distribution, Resistance profiles, Genomic/phylogenetic analysis
Answer: We appreciate your insight, for this reason, we have modified the Results following your advice to improve the clarity. Moreover, we have removed our incidental finding such as MRSA, according to another reviewer’s suggestion.
- Tables 1 and 2 provide extensive genomic data on E coli isolates from environmental and human sources. These tables are comprehensive and technically valuable. However, despite the depth of information, the tables are not fully utilized in the main text. The Results sections that reference them (2.3 and 2.4) tend to reiterate individual data points (e.g., "13 different STs were identified") without drawing meaningful synthesis or interpreting broader trends—such as which STs are shared across environmental and clinical settings, or which plasmid types are most strongly associated with multidrug resistance. Furthermore, the lack of summary statistics in the text (e.g., percentages of isolates carrying major ARGs or replicon families) forces readers to manually scan the tables to extract key insights. This weakens the tables' impact and makes it harder to assess patterns.
Answer: Thank you for your comments. In the results section, we have improved the information to make it clearer and more accurate, both in the tables and in the text.
- There is a Lack of Integration Between Environmental and Human Finding. Although Section 2.4 covers rectal swab results, it feels disconnected from the environmental findings. Add a transitional paragraph or summary at the end of Section 2 summarizing similarities and suggesting possible links (currently deferred until the Discussion).
Answer: A concluding paragraph has been added to the section to discuss the observed similarities and potential epidemiological links between the isolates.
- The manuscript does not specify which antibiotics were used to assess multidrug resistance, nor does it report MIC values or susceptibility profiles beyond genetic data. This is a significant omission. The interpretation of multidrug resistance should be supported by phenotypic susceptibility testing, ideally referencing EUCAST/CLSI breakpoints. A brief table or supplementary data listing tested antibiotics and resistance rates would improve transparency.
Answer: We apologise for this oversight and have added this data into the methodology and designed a table with the resistance percentages of the antibiotics tested in ESBL-producing E. coli.
- While WGS was performed on select isolates, the data remains underutilized and largely descriptive. To fully leverage the potential of this data and improve the impact of the study, additional genomic analyses should be considered. Such as- mobile genetic element analysis to identify integrons, transposons, or insertion sequences that may facilitate the transfer of resistance genes, and plasmid reconstruction to assess the co-localization of multiple resistance determinants on the same plasmid. A pan-genome analysis could reveal gene gain or loss events and functional differences between isolates from environmental and human sources. Furthermore, comparative resistome profiling using clustering or ordination methods (e.g., PCA or heatmaps) would help highlight similarities or divergence across sample types.
Answer: The reviewer is correct; however, due to limitations in bioinformatics resources and financial support, we were unable to perform the proposed analysis in the current study. Nonetheless, we greatly appreciate the suggestion and will consider it for future research.
- Additionally, qPCR-based quantification of key ARGs could substantially enhance the scientific depth of the manuscript.
Answer: Unfortunately, we could not to carry out the analysis that you propose in this study.
- Figure 2 does not indicate the number of isolates (n) per species. The caption should also include these counts.
Answer: We have added the number of isolates as a footnote following your advice.
- All microbial names should be italicized. After first use, only the first letter of the genus should be used.
Answer: Thank you for your suggestion. We have revised and modified it accordingly.
Round 2
Reviewer 1 Report
Comments and Suggestions for Authors
All the recommended corrections have been appropriately addressed by the authors.
Reviewer 2 Report
Comments and Suggestions for Authors
The authors have addressed all the recommendations from the previous review and have revised the manuscript accordingly. The manuscript now fully meets the scientific and technical standards of the journal. The topic is highly relevant and timely, particularly in the context of the One Health approach to antimicrobial resistance surveillance and control.
Reviewer 4 Report
Comments and Suggestions for Authors
The authors have addressed all the concerns